# Radiotherapy for Clinically Localized T3b or T4 Very-High-Risk Prostate Cancer-Role of Dose Escalation Using High-Dose-Rate Brachytherapy Boost or High Dose Intensity Modulated Radiotherapy

**DOI:** 10.3390/cancers13081856

**Published:** 2021-04-13

**Authors:** Hideya Yamazaki, Gen Suzuki, Koji Masui, Norihiro Aibe, Daisuke Shimizu, Takuya Kimoto, Ken Yoshida, Satoaki Nakamura, Haruumi Okabe

**Affiliations:** 1Department of Radiology, Graduate School of Medical Science, Kyoto Prefectural University of Medicine, 465 Kajiicho Kawaramachi Hirokoji, Kamigyo-ku, Kyoto 602-8566, Japan; gensuzu@koto.kpu-m.ac.jp (G.S.); mc0515kj@koto.kpu-m.ac.jp (K.M.); a-ib-n24@koto.kpu-m.ac.jp (N.A.); dshimizu@koto.kpu-m.ac.jp (D.S.); t-kimoto@koto.kpu-m.ac.jp (T.K.); 2Department of Radiology, Kansai Medical University, Hirakata 573-1010, Japan; yoshidaisbt@gmail.com (K.Y.); satoaki@nakamura.pro (S.N.); 3Department of Radiology, Ujitakeda Hospital, Uji-City, Kyoto 611-0021, Japan; h-okabe@takedahp.or.jp

**Keywords:** prostate cancer, T3b–T4, very high risk, high dose rate, brachytherapy, IMRT

## Abstract

**Simple Summary:**

Recently, high-risk prostate cancer was subdivided to a very-high-risk group considered to have the worst prognosis, including clinical stage T3b–T4, primary Gleason pattern 5, or more than four biopsy cores with Gleason score 8–10. Among these, T3b–T4 stage is a special interest in radiotherapy because of their wider target volume outside the prostate. We examined this subgroup and found that dose escalation in radiotherapy both with brachytherapy or intensity modulated radiotherapy (IMRT) improved biochemical free survival rate but not in prostate cancer specific survival rate and overall survival rate.

**Abstract:**

To examine the efficacy of dose escalating radiotherapy into patients with cT3b or T4 localized prostate cancer, we compared Group A (86 conventional dose external beam radiotherapy: EBRT group, treated with 70–72 Gy) and group B (39 high dose EBRT group (HDEBRT group, 74–80 Gy) and 124 high-dose-rate brachytherapy (HDR) + EBRT (HDR boost)) using multi-institutional retrospective data. The actuarial 5-year biochemical disease-free survival (bDFS) rate, prostate cancer specific survival rate (PSS), and overall survival rate (OS) were 75.8%, 96.8%, and 93.5%. Group B showed superior 5-year bDFS rate (81.2%) as compared to the group A (66.5%) (*p* < 0.0001) with a hazard ratio of 0.397. Equivocal 5-year PSS (98.3% and 94.8% in group B and group A) and OS (both 93.7%) were found between those groups. Accumulated late grade ≥ 2 toxicities in gastrointestinal and genitourinary tracts were similar among those three groups. Therefore, both HDEBRT and HDR boost could be good options for improving the bDFS rate in cT3–T4 localized prostate cancer without affecting PSS and OS.

## 1. Introduction

Prostate cancer is the most common solid organ malignancy in men in developed countries [1], and the second leading cause of cancer death in the United States of America (USA) [1]. Although advanced treatments have improved outcome, it has become difficult to choose the best treatment modality because there are many curative treatment options, such as surgery, external beam radiotherapy (EBRT), and brachytherapy (BT) [2]. In addition, high-risk prostate cancer was recently subdivided to include a very-high-risk group considered to have the worst prognosis, including clinical stage T3b–T4, primary Gleason pattern 5, or more than four biopsy cores with Gleason score 8–10 [2]. Among these, the T3b–T4 stage is an important issue in radiotherapy because of their wider target volume outside the prostate [3]. However, there is a lack of literature for outcomes of the T3b–4 very-high-risk group after radiotherapy [4,5].

Many randomized controlled trials have demonstrated the superiority of treatment with increasing prescribed dose for localized prostate cancer [6] in biochemical disease free (bDFS) rate. Consequently, the National Comprehensive Cancer Network (NCCN) Clinical Practice Guidelines in Oncology (2019) state that doses of 70 Gy in conventional fractions delivered to the prostate are not appropriate for patients with localized prostate cancer [2]. Therefore, we compared the outcome of conventional (Conv) EBRT using the prescribed dose of 70–72 Gy to high dose EBRT using intensity modulated radiotherapy (IMRT) prescribed dose 74 Gy or more and to high-dose rate brachytherapy with EBRT (HDR boost).

HDR was used alone or combined with EBRT (HDR boost) and good efficacy was obtained in all risk groups [7,8,9,10]. We previously compared low-dose-rate brachytherapy ± EBRT and HDR [9], but we did not investigate the role of brachytherapy in T3b–4 because the low dose rate (LDR)-BT was not used to treat advanced disease. With modern imaging and techniques, we could speculate that HDR could potentially treat the T3b–4 diseases that had invaded outside the prostate [10], and we examined the outcome of HDR boost in cT3b–4 prostate cancer.

To analyze a large cohort, we used freely available public data regarding HDR boost and EBRT [11], including dose escalating IMRT performed in our institution [12]. Therefore, the aim of the present study was to examine the efficacy of radiotherapy in clinically localized T3b–4 N0M0 prostate cancer and to evaluate the role of dose escalation using HDR-BT boost or high dose IMRT.

## 2. Materials and Methods

### 2.1. Patients

We examined the data of patients treated with HDR boost (open data for public use) [11] and EBRT (open data and Uji-Takeda Hospital) in a retrospective fashion (Table 1). Patient eligibility criteria included: treatment with HDR boost or EBRT, clinical stage T3b–T4, N0M0; All tumor was histology-proven adenocarcinoma, and pretreatment PSA level (initial prostate specific antigen = iPSA) level, Gleason score sum (GS), and T classification are available. The patients were staged according to the NCCN risk classification [2]. We compared group A (86 conventional dose external beam radiotherapy: EBRT group = control group) and group B (39 high dose EBRT group (HDEBRT group, 74–80 Gy) and 124 high-dose-rate brachytherapy (HDR) + EBRT (HDR boost)). The Phoenix definition (nadir, +2 ng/mL) was applied to prostate specific antigen (PSA) failure. Toxicity analysis was performed using The Common Terminology Criteria for Adverse Events version 4.0. All patients in the Uji-Takeda group provided written informed consent and patients in public data gave their informed consent during the process of building public data. This study was conducted under permission of institutional review board permission (Kyoto Prefectural University of Medicine: ERB-C-1403), and in accordance with the Declaration of Helsinki. In general, patients were followed up at 3-month intervals during the first year and at 3–6-month intervals thereafter [10,12].

### 2.2. Treatment Planning

#### 2.2.1. High Dose Rate Brachytherapy Boost with External Beam Radiotherapy (HDR Boost)

The multi-institution data was obtained from open data source [11], and detailed method of the applicator implantation was described elsewhere [10]. All patients were treated with a combination of HDR and EBRT at various fractionations (Table 2). The median dose of HDR used was 31.5 Gy (18–31.5 Gy) and that of EBRT was 39 Gy (39–51 Gy). The median fraction size of HDR was 6.3 Gy (6.3–10.5 Gy) and that of EBRT was 3 Gy (2–3 Gy).

#### 2.2.2. External Beam Radiotherapy (EBRT: Conv. and High Dose Group)

The EBRT group consisted of Conv EBRT and high dose EBRT (HDEBRT) groups; Conv. EBRT Group included the treatment group whose prescribed dose was ≤72 Gy in EQD 2Gy, and HDRT included treatment prescribed ≥74 Gy using IMRT. Conv EBRT group used several modalities (conventional two-dimensional treatment planning (2D), three-dimensional conformal radiotherapy (3D-CRT) and IMRT, and details are shown in Table 2. EBRT data were partly obtained from freely accessible public dataset (*n* = 225) [11]. Image-guided IMRT with helical tomotherapy was performed at the Department of Radiology, Ujitakeda Hospital (*n* = 20). The detailed technique of image-guided IMRT with helical tomotherapy has been described elsewhere [12]. In Ujitakeda hospital, we used volume dose prescriptions for D95 (95% of planning target volume (PTV) received at least prescribed dose) of 74.8 Gy/34 fractions (2.2 Gy/fraction, *n* = 8) between June 2007 to 2009. We changed the prescribed dose to 74 Gy/37 fractions (2 Gy/fraction, *n* = 12) from June 2009 to September 2013 [12].

### 2.3. Statistical Analysis

StatView 5.0 statistical software was used for statistical analyses. Chi-square tests and Student’s *t*-tests were used for percentages. Means or medians were compared with Mann–Whitney U-tests (two variables) and Kruskal–Wallis test (three variables). To analyze the biochemical control rate, survival, and accumulated toxicity, we used the Kaplan–Meier method, and compared them by log-rank tests. Univariate and multivariate analyses were made with Cox’s proportional hazard model. *p* < 0.05 was considered statistically significant.

## 3. Results

### 3.1. Patient and Tumor Characteristics

The 249 patients with stage T3b–T4 N0M0 prostate cancer were treated with HDR boost (*n* = 124) or EBRT (Conv EBRT *n* = 86, and HDEBRT *n* = 39). The median patient age was 71 (range, 60–89) years. The patients’ clinical characteristics are shown in Table 1 and Table 2. Detailed comparison among three groups (Conv EBRT vs. HDR boost vs. HDEBRT), and BED/EQD2Gy for each treatment were shown in Appendix A. The median follow-up duration for the entire cohort was 64 (range: 13–153) months, with a minimum of 2 years for surviving patients or until death. EBRT was used to treat patients with advanced disease (T category) with a shorter period of hormonal therapy and lower prescribed dose than that in the HDR boost group.

### 3.2. Biochemical Control (bDFS), Prostate Cancer-Specific (PSS), and Overall (OS)

Of the total, the actuarial 5-year biochemical control (5y-bDFS) rate was 75.8% (95% confidence interval (CI): 69.7–81.9%) (Figure 1) at 5 years, and 56.4% (47.0–65.9%) at 10 years. The actuarial prostate cancer specific survival rate (PSS) was 96.8% (95% CI: 94.3–99.4%) at 5 years and 90.9% (84.2–97.6%) at 10 years (Figure 1). The overall survival rate was 96.8% (95% CI: 94.3–99.4%) at 5 years and 90.9% (84.2–97.6) at 10 years (Figure 1).

Group A showed bDFS rates of 66.5% (95 CI; 56.1–76.9%) at 5 years and 38.0% (24.2–51.7%) at 10 years; Group B 81.2% (73.8–88.6) at 5 years 71.3% (60.3–82.2) at 10 years (*p* < 0.0001, Figure 2).

Conv EBRT group showed bDFS rates of 66.5% (95 CI; 56.1–76.9%) at 5 years and 38.0% (24.2–51.7%) at 10 years; HDR boost group 78.9%, (69.8–87.9) at 5 years and 67.7% (55.5–80.0) at 10 years; and HDEBRT group 88.1% (77.2–99.2%) at 5 years. There was a statistically significant difference among those three groups (*p* = 0.0003) (Figure 3).

The corresponding bDFS rate was 77.6% (95% CI = 71.2–84%) at 5 years and 62.1% (52.7–71.6%) at 10 years in the T3b group (*p* = 0.0548) and 63.5% (45.0–82.1%) at 5 years and 26.4% (29–49.9%) at 10 years in the T4 group (*p* = 0.0060 between T3b and T4).

In a detailed analysis of T3b tumor at 5 and 10 years, Conv EBRT showed bDFS rates of 69.9% (58.4–81.3%) and 44.0% (28.1–59.8), respectively, and HDR boost of 79.9% (70.6–89.1%) and 71.3% (59.8–82.8%), respectively; HDEBRT showed a bDFS rate of 86.7% (74.4–99.0%) at 5 years (*p* = 0.0084 among those 3 groups). For T4 tumor, Conv EBRT showed 5y-bDFS rate of 53.8% (30–77.7%) and was the same until 111 months, whereas HDR boost had a 5y-DFS of 68.6% (32.1–100%) at 5 years, 34.3% (16.6–85.2%) at 10 years, and HDEBRT 100% at 5 years (*p* = 0.0171 for 100% of HDEBRT among those 3 groups).

As shown in Table 3, the predictors of biochemical control on univariate analysis included treatment modality (Group A vs. Group B), Gleason score (≤7 vs. 8≤), and pretreatment PSA level (≤20 vs. >20 ng/mL). On multivariate Cox regression analysis, the Gleason score (Hazard ratio (HR) = 2.004, 95% CI = 1.182-3.399, *p* = 0.0099), pretreatment PSA (HR = 1.984, 95% CI = 1097–3.215, *p* = 0.0215), and treatment arm (HR = 0.397, 95% CI = 0.241–0.654, *p* = 0.0003) remained significant for influencing biochemical control.

Group A showed PSS rates of 94.8% (95 CI; 89.9-99.8%) at 5 years and 88.1% (79.4–79.6%) at 10 years; Group B 98.3% (95.8–1.007) at 5 years 92.8% (82.2–1.035) at 10 years (*p* = 0.0883, Figure 2). Conv EBRT showed 94.8% PSS (89.9–99.8%) at 5 years and 88.1% (79.4–96.9) at 10 years, whereas HDR boost 97.7% (94.4–100%) at 5 years and 92.3% (81.5–100%) at 10 years, and HDEBRT 100% at 5 years. There were no statistically significant differences among the three groups (*p* = 0.189) (Figure 3).

An in depth analysis of T3b tumor revealed that conv EBRT showed 95.3% PSS (90–100%) at 5 years and 89.9% (81.1–98.7%%) at 10 years, whereas HDR boost 99.1% (97.3–100%) at 5 years and 93.3% (92.1–104%) at 10 years, and HDEBRT 100% at 5 years until 87.7 months (*p* = 0.162 among those 3 groups). For T4 tumor, conv EBRT showed 5y-PSS 93.3% (80.7–106%) at 5 years and 81.7% (57.6–105%) at 10 years, whereas HDR boost 80% (44.9–115%) at 5 years, and until 153 months (*p* = 0.801 among those 3 groups).

Group A showed OS rates of 93.7% (95 CI; 89.6–97.8%) at 5 years and 84.0% (72.6–95.3%) at 10 years; Group B 93.7% (89.6–97.8%) at 5 years 84.0% (72.6–95.3%) at 10 years (*p* = 0.8570, Figure 3). Conv EBRT showed overall survival rate 93.7% (95 CI; 88.3–99.1%) at 5 years and 82.1% (71.3–92.9%) at 10 years, whereas HDR boost 91.8%, (86.5–97.0%) at 5 years and 83.6% (72.2–94.6%) at 10 years, and 100% at 5 years and 80% (44.9–100%) at 87.7 months in HDEBRT groups. There were no statistically significant differences among the three groups (*p* = 0.5311) (Figure 4).

The corresponding PSS rates were 93.9% at 5 years, 83.0% at 10 years in the T3b group and 90.9% at 5 years and 81.8% at 10 years in the T4 group, (*p* = 0.9079 between T3b and T4).

For T3b tumors, conv EBRT showed 93.8% (87.8–99.7%) PSS at 5 years and 82.1% (69.9-94.2%) at 10 years, whereas HDR boost showed 92.6% (787.6–96.8%) at 5 years and 83.8% (71.9–95.6%) at 10 years, and HDEBRT showed 100% at 5 years until 87.7 months (*p* = 0.5894 among those 3 groups). For T4 tumor, conv EBRT showed 5y PSS of 93.3% (56.1–76.9%) and 10y PSS of 81.7% (57.6–1.057%), whereas HDR boost 80% (44.9–1.151) at 5 years, and HDEBRT 100% (1-1) at 5 years until 90 months (*p* = 0.801 among those 3 groups).

### 3.3. Toxicity

#### 3.3.1. Acute Toxicity

Table 4 shows maximal toxicity grade after radiotherapy according to each group. Higher but mild acute gastrointestinal (GI) toxicity was found in group A than in group B (grade ≥ 1 toxicity: 48% vs. 8%, *p* < 0.0001) groups. The incidence of genitourinary (GU) toxicity was slightly higher trend in the group B than group A (grade ≥ 2: 16% vs. 6%, *p* = 0.056). Appendix A shows the incidence of late GI and GU toxicities among three groups. Higher but mild acute gastrointestinal (GI) toxicity was found in Conv EBRT group than in HDEBRT and HDR boost (grade ≥ 1 toxicity: 48% vs. 5% and 9%, *p* < 0.0001) groups. The incidence of genitourinary (GU) toxicity was higher in the HDR boost group next to HD EBRT and conv EBRT (grade ≥ 2: 18% vs. 6% 10%, *p* < 0.0001). However, grade 3 toxicities or more were not observed.

#### 3.3.2. Late Toxicity

Table 5 shows the incidence of late GI and GU toxicities between Group A and B. Higher but mild acute gastrointestinal (GI) toxicity was found in group A than in group B (grade ≥ 1 toxicity: 23% vs. 12%, *p* = 0.0382) groups. The incidence of mild genitourinary (GU) toxicity was higher in the group B than group A (grade ≥ 1: 44% vs. 20%, *p* = 0.0004); however, grade 4 toxicities were not observed.

Appendix A shows the incidence of late GI and GU toxicities among three groups. An elevated incidence of GU toxicities was observed in the Conv EBRT and HDR boost groups than in HDEBRT group (grade 1≥ toxicity: 62%, 71%, 33%, grade ≥ 2 toxicity: 6% and 18% vs. 10%, *p* < 0.0001); however, grade 4 toxicities were not observed.

The accumulated incidence rates of grade ≥2 GI toxicities were 1.3% (0–3%) at 5 years and 1.3% (0–3%) at 10 years in group B, whereas it was 3.6% (0–7.5%) and 3.6% (0–7.5%) in group A (*p* = 0.2687, Figure 4).

Those figures were 3.6% (95% CI: 0-7.5%), 5.1% (0–12.1%), and 1.0% (0–3.1%) in the Conv EBRT, HD EBRT, and HDR boost groups (Figure 3, *p* = 0.2436) at 5 years, respectively (Figure 5).

The accumulated incidence rates of grade ≥2 GU toxicities were 9.3% (4.5–14.2%) at 5 years and 23.2% (5.9–40.4%) at 10 years for group B, whereas 9.2% (2.7–15.7%) and 13.8% (4.8–22.7%) for group A (*p* = 0.5268, Figure 4).

Those figures were 9.2% (95% CI: 2.7–15.7%), 5.6% (0–13.2%), and 11.4% (5.3–11.6%) in the Conv EBRT, HD EBRT, and HDR boost groups at 5 years (Figure 5, *p* = 0.5120), respectively. Actuarial late grade ≥2 toxicities in GI and GU were similar among those three groups.

## 4. Discussion

High-risk prostate cancer was subdivided into a very high risk group to include clinical stage T3b–T4 lesions, primary Gleason pattern 5, or more than four biopsy cores with Gleason score 8–10 [2]. Of these, T3b–T4 factor obtained a special focus in radiotherapy for target delineation and treatment choice [3]. Herein, we presented evidence of efficacy of elevated dose radiotherapy (HDR boost and HDEBRT) in clinical localized T3b–T4 prostate cancer using comparison analysis. For very high risk group, Narang et al. reported 46% of 10-year bDFS including 82% of 3D-CRT and 18% of IMRT with prescribed dose 70.2 Gy (64.8–75.6 Gy) [13]. Goupy et al. reported a 5-years bDFS rate of 75.2% in T3bN0 disease with 74 (70–76) Gy of IMRT (FU 26 months) in French population [14]. Ishiyama reported 5-year bDFS of 81.9% using HDR + EBRT [4]. Our data (Conv EBRT 66.5% and 38.0%, HDR 78.9% and 67.7%, and HDEBRT 88.1% and not available at 5 and 10-years) seems similar.

A number of studies provide evidence for the efficacy of dose escalation in prostate cancer [2,14,15,16,17,18,19,20,21]. Our results are in line with previous findings that suggest that elevated dose could improve the bDFS rate in almost all risk groups [2,14,15,16,17,18,19,20,21]. We thought there were three important points to improve outcome; combination of high-dose, high-precision (IMRT and image guided radiotherapy; IGRT) radiotherapy, and androgen deprivation therapy (ADT). At first, high dose radiotherapy improved outcome [2,14,15,16,17,18,19,20]. For example, Eade et al. recommended doses of > or =80 Gy for most men with prostate cancer (5y bDFS <70 Gy, 70–74.9 Gy, 75–79.9 Gy, and >80 Gy = 70%, 81%, 83%, and 89%) [17], and Pollack et al. confirmed 78 Gy arm showed superior outcome than 70 Gy arm (bDFS rates for the 70 and 78 Gy arms at 6 years were 64% and 70%) [18]. Our threshold between Conv EBRT and HDEBRT seems lower than those of reported high dose arm. Next, high-precision (IMRT and IGRT) radiotherapy could play an important role, enabling to deliver higher dose without increasing toxicity. Simultaneously, we changed dose prescription method from point dose prescription (reference point, isocenter) in 3D-CRT into volume dose prescription (i.e., D95) for IMRT using IGRT. Therefore, 74 Gy dose volume prescription methods actually prescribed a dose 105–110% higher (77.7–81.4 Gy) than point dose prescription. As for the role of ADT, Zapatero et al. has shown a 5-year biochemical disease-free survival of 63% for dose less than 72 Gy versus 84% for dose of at least 72 Gy in the cohort of high-risk patients treated with 3D-CRT with neoadjuvant and 2-year adjuvant ADT after EBRT [20]. We used long term ADT, which could be one of the reasons of our good outcome compared to previous studies. Furthermore, the good efficiency of ADT was found in Japanese men and is explained by the Japanese-specific high-sensitivity to hormonal therapy [22].

Of all techniques, BT has a unique character that allows it to deliver higher doses of radiation to the target lesion without excessive irradiation of the adjacent organs, and is considered to be one of the best radiotherapy options [23]. In addition, the lowα/β ratio and consequent high sensitivity to dose fractionation of prostate adenocarcinoma could be translated to higher sensitivity to large radiation doses per fraction than for most other malignancies [2,14]. Therefore, we hypothesized that the use of hypo-fractionated schedules could be a good strategy to achieve better outcomes with dose escalation; this is optimally achieved with HDR [24]. Several retrospective studies and a few preliminary and premature prospective studies have reported benefits of HDR [7,24]. However, these randomized control trials majorly treated low and intermediate group prostate cancers and did not include the very-high-risk group [24]; therefore, providing room for investigation of very-high-risk groups, such as in T3b and T4 diseases. Our data could provide useful insight into daily clinical decisions in this field. We also investigated HDR monotherapy and reported 5y-bDFS rate of 86.4% in T3bT4 disease (*n* = 37; 89.1%, and 77.8% for T3b and T4, respectively) [25], indicating that HDR monotherapy was also a promising procedure to achieve a good outcome compared to Conv EBRT.

Some authors reported superiority of HDR boost over HD EBRT in the bDFS rate [26]. Spratt et al. reported that enhanced dose escalation using combo-RT was associated with superior bDFS outcomes for patients with intermediate-risk prostate cancer compared with high-dose IMRT alone even at a dose of 86.4 Gy, but not for the high-risk group [26]. Furthermore, several studies found superior efficacy of dose escalation not only in terms of bDFS, but also PSS and OS [27,28]. On the other hand, our data indicated that dose escalation did not translate into improving PSS or OS. Recent advances in treatments have improved outcomes. Ngnenn et al. also reported excellent outcome of high-risk prostate cancer with modern, high-dose, EBRT, and androgen-deprivation therapy (ADT) and the strategy could produce better biochemical, clinical, and survival outcomes than those from previous eras [29]. Specifically, symptomatic local failure was uncommon, and few men died of prostate cancer even 10 or more years after treatment [21]. Actually, the age adjusted PSS of clinically localized prostate cancer treatment has reached nearly 100% in Japan at 10 years [30] and at 5 years in USA [1].

For GU toxicity, Carvalho, et al. made a systematic review and meta-analysis and reported around 28% of late GU toxicity at 12 months or more after EBRT [31]. Takemoto reported 7.9–12.4% of GU Grade ≥2 ratio at 6–10 years using IMRT in Japan [32]. For HDR+EBRT, Ishiyama et al. reported 5- and 10-year accumulated rates of late Grade ≥2 GU toxicities were 16.7% and 26.7% in 3424 Asian patients [10]. Our data (Conv EBRT 9.2% and 13.8%, HDEBRT 5.6% and not available, HDR +EBRT 10.5%, and 23% at 5- and 10-years) concurred to Japanese and Asian data and it seems lower than the previous review [31].

We admitted to several limitations of the present study, including its retrospective nature, limited follow-up time, and small sample size, especially in the HD EBRT group, which may limit the application of its findings. Thus, a longer follow-up with a larger sample is needed for reaching concrete conclusions. Randomized, prospective studies are needed to confirm these findings. Furthermore, as longer use of ADT could mask the radiotherapy efficacy, the follow-up period of 64 months may be short to fully assess the outcomes. Finally, although using a free database is beneficial, retrospective databases may not record toxicity outcomes properly. Although our data did not reveal difference among the three groups in terms of the toxicity profile, BT has showed lower GI toxicity and higher GU toxicity than EBRT in general [7,23], and HDEBRT (IMRT) using modern image guided radiotherapy technique could reduce GI toxicity with meticulous technical consideration [2,11].

## 5. Conclusions

This study showed that HDR boost and high dose EBRT improved bDFS more than Conv EBRT in clinical T3b and T4 prostate cancer patients, with equivocal PSS, OS, and late toxicity profile.

## Figures and Tables

**Figure 1 cancers-13-01856-f001:**
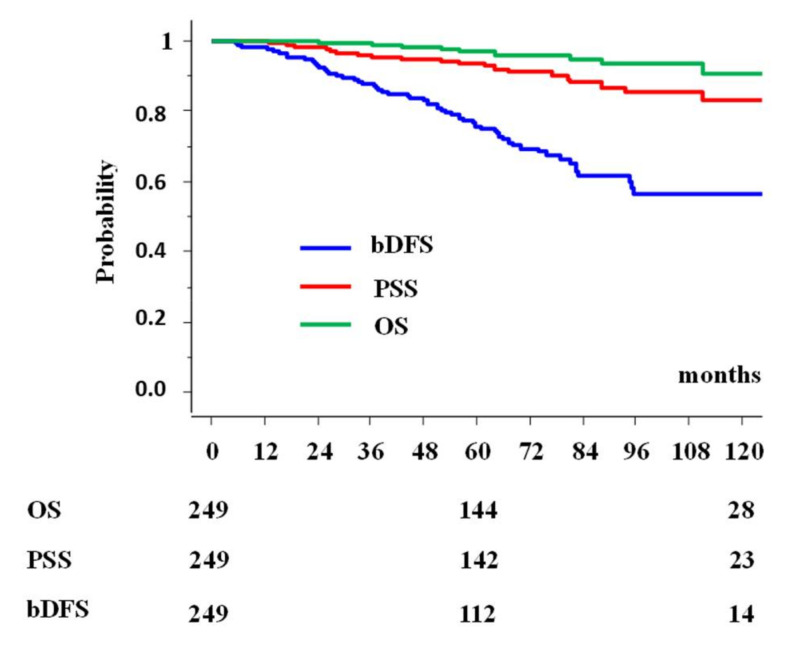
Biochemical disease-free rate (bDFS), prostate cancer specific survival rate (PSS), and overall survival rate (OS) in clinically T3b–T4 prostate cancer.

**Figure 2 cancers-13-01856-f002:**
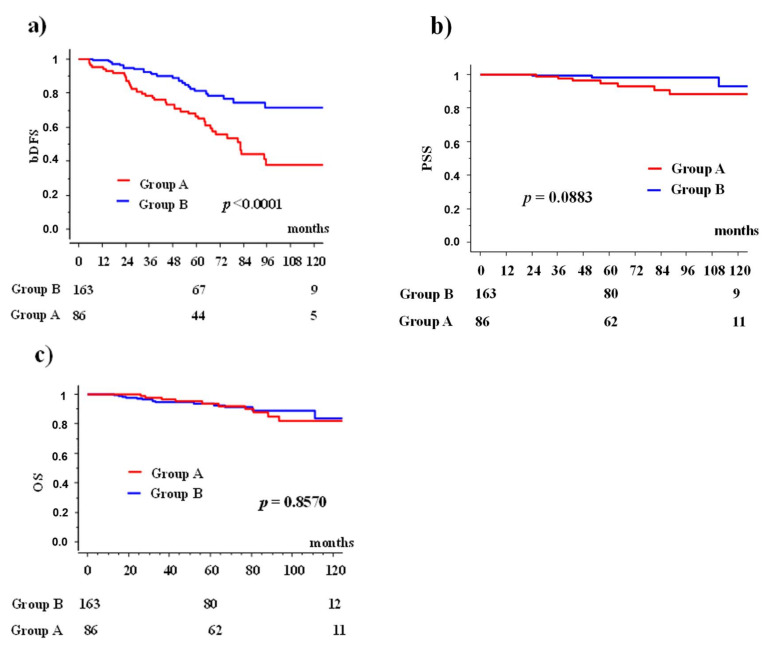
Biochemical disease-free rate (bDFS), prostate cancer specific survival rate (PSS), and overall survival rate between group A and B; (**a**) biochemical control rates; (**b**) prostate cancer specific survival rate; and (**c**) overall survival rate.

**Figure 3 cancers-13-01856-f003:**
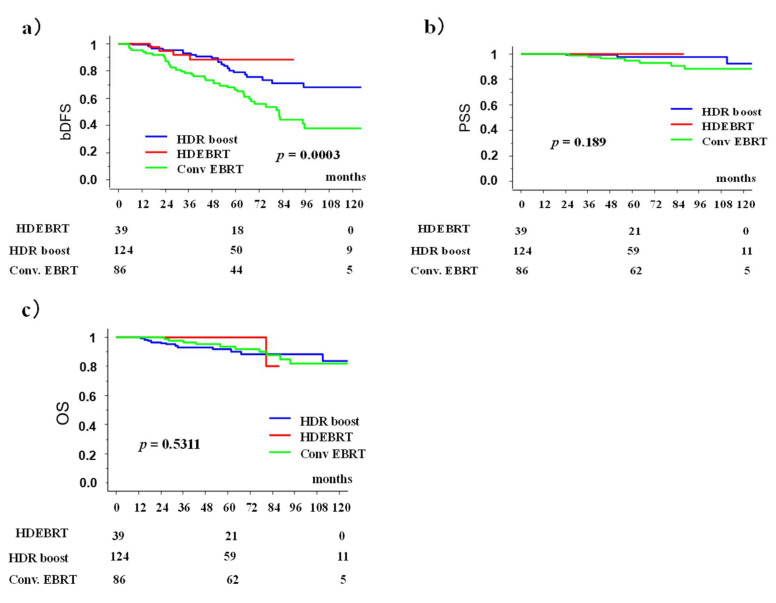
Biochemical disease-free rate (bDFS), prostate cancer specific survival rate (PSS), and overall survival rate (OS) among conventional external beam radiotherapy (Conv EBRT), high dose external beam radiotherapy (HDEBRT), and high dose rate brachytherapy (HDR) boost groups; (**a**) biochemical control rates; (**b**) prostate cancer specific survival rate; and (**c**) overall survival rate.

**Figure 4 cancers-13-01856-f004:**
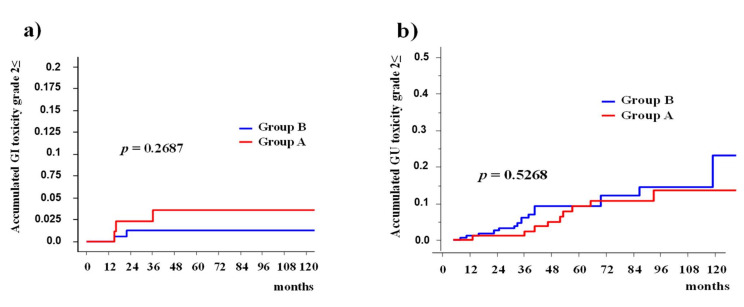
Accumulated incidence of grade ≥ 2 late toxicity between Group A and B. (**a**) Accumulated incidence of grade ≥ 2 gastrointestinal (GI). (**b**) Accumulated incidence of grade ≥ 2 genitourinary (GU) toxicity.

**Figure 5 cancers-13-01856-f005:**
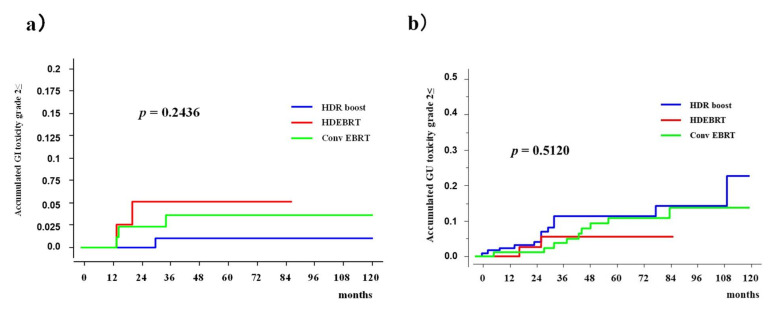
Accumulated incidence of grade ≥2 late toxicity among Conv EBRT, HDEBRT, and HDR boost groups. (**a**) Accumulated incidence of grade ≥2 gastrointestinal (GI) toxicity. (**b**) Accumulated incidence of grade ≥2 genitourinary (GU) toxicity.

**Table 1 cancers-13-01856-t001:** Comparison of patients Characteristics between group A and group B.

Variables	Strata	Group A		Group B		*p*-Value
		Conv. EBRT		High Dose EBRT + HDR Boost (*n* = 163)		
		(*n* = 86)				
		No. Or	(%)	No. Or	(%)	
		Median (Range)		Median (Range)		
Age		70.5 (60–89)		72 (60–86)		0.3999
T category	3b	67	−78%	149	−91%	
	4	19	−22%	14	−9%	**0.0028**
iPSA	(ng/mL)	42.07 (3.4–398)		31.5 (4.7–486)		0.5272
Gleason score	−6	0	0%	13	−8%	
	7	34	−40%	64	−39%	
	8	51	−59%	86	−53%	**0.0258**
Prescribed dose	(Gy)	72 (70–72)		108.7 (74–129)		**<0.0001**
(EQD 2 Gy)						
Hormonal therapy	Yes	86	−100%	159	−98%	0.349
Neoadjuvant	months	7.5 (2–18)		11 (11–74)		**<0.0001**
Adjuvant	months	20 (1–30)		36 (3–114)		**<0.0001**
	No	0	0%	4	−2%	
Follow-up	(Months)	77.5 (18.7–135)		57 (13–153)		**<0.0001**

Bold values indicate statistically significance, NA; not available. HDR boost = high dose rate brachytherapy boost, EBRT = external beam radiotherapy. EQD 2 Gy = *n* × d × (α/β + d)/(α/β + 2) (α/β = 1.5 Gy, *n* = fraction number, d = single dose).

**Table 2 cancers-13-01856-t002:** Treatment schedule.

Group A			Group B					
Conv. EBRT (*n* = 86)			High Dose EBRT (*n* = 39)			HDR Boost (*n* = 124)		
Prescribed Dose	No.	(%)	Prescribed Dose	No.	(%)	Prescribed Dose	No.	(%)
70 Gy/35 fr ^1^	14	−16%	74 Gy/36 fr	12	−31%	18 Gy/2 fr +EBRT 39 Gy/13 fr (*n* = 10) or	25	−20%
						51 Gy/17 fr (*n* = 14) or		
						48 Gy/16 fr (*n* = 1)		
72 Gy/36 fr ^2^	72	−84%	78 Gy/39 fr ^3^	14	−36%	20 Gy/2 fr + EBRT46 Gy/23 fr ^4^	4	−3%
			74.8 Gy/34 fr	8	−21%	21 Gy/2 fr + EBRT 45 Gy/15 fr (*n* = 4)	5	−4%
						21 Gy/3 fr + EBRT 51 Gy/17 fr (*n* = 1)		
			80 Gy/40 fr	4	−10%	31.5 Gy/5 fr + EBRT 30 Gy/10 fr	90	−73%
			70 Gy/28 fr	1	−3%			

Bold values indicate statistically significance, NA; not available, HDR boost; high dose rate brachytherapy boost, EBRT; external beam radiotherapy, IMRT; intensity modulated radiotherapy. ^1^ three-dimensional conformal radiotherapy; 3CDRT *n* = 13, intensity modulated radiotherapy; IMRT *n* = 1. ^2^ two-dimensional radiotherapy; 2D + 3D-CRT *n* = 33, 3D-CRT *n* = 13, IMRT *n* = 14, IMRT+3D-CRT *n* = 12 (2D + 3D-CRT included Whole pelvis 40 Gy/20fr *n* = 32, 46 Gy/23fr *n* = 1, and IMRT Whole pelvis 46 Gy/23fr *n* = 3). ^3^ included whole pelvis radiotherapy 46 Gy *n* = 7. ^4^ IMRT *n* = 4.

**Table 3 cancers-13-01856-t003:** Univariate and multivariate analysis for biochemical control rate using Cox proportional hazards model.

Variables	Strata	Univariate Analysis			Multivariate Analysis		
		HR	95% CI	*p*-Value	HR	95% CI	*p*-Value
**Age, years**	≤70	1	(referent)	-	1	(referent)	-
	71≤	1.498	0.925–2.425	0.1004	1.67	1.007–2.769	**0.0468**
**Gleason score**	≤7	1	(referent)	-	1	(referent)	-
	8≤	2.148	1.270–3.634	0.0043	2.004	1.182–3.399	**0.0099**
**Pretreatment PSA (ng/mL)**	≤20	1	(referent)	-	1	(referent)	-
	20<	1.424	0.863–2.351	0.167	1.984	1.097–3.215	**0.0215**
**Modality**	Group A	1	(referent)	-	1	(referent)	-
	Group B	0.258	0.092–0.723	0.0011	0.397	0.241–0.654	**0.0003**

Bold values indicate statistically significance. Abbreviations; CI = confidence interval; HR = hazard ratio, NA = not available. Group A = Conv. EBRT, Group B = HDR boost + HDEBRT.

**Table 4 cancers-13-01856-t004:** Acute toxicity between Group A and B.

Toxicities	Grade	Group A. (*n* = 86)		Group B (*n* = 163)		*p*-Value
		No.	(%)	No.	(%)	
**Genitourinary**	0	33	−38%	62	−38%	**0.0566**
	1	48	−56%	75	−46%	
	2	5	−6%	26	−16%	
	3	0	0%	0	0%	
**Gastrointestinal**	0	45	−52%	150	−92%	**<0.0001**
	1	40	−47%	13	−8%	
	2	1	−1%	0	0%	
	3	0	0%	0	0%	

Bold values indicate statistically significance, HR; hazard ratio, CI; confidence interval. HDR boost = high dose rate brachytherapy boost, EBRT = external beam radiotherapy.

**Table 5 cancers-13-01856-t005:** Late toxicity between Group A and B.

Toxicities	Grade	Group A. (*n* = 86)		Group B (*n* = 163)		*p*-Value
		No.	(%)	No.	(%)	
**Genitourinary**	0	69	(80%)	92	(56%)	**0.0004**
	1	8	(9%)	55	(34%)	
	2	5	(6%)	9	(6%)	
	3	4	(5%)	7	(4%)	
**Gastrointestinal**	0	65	(76%)	145	(89%)	**0.0382**
	1	18	(21%)	14	(9%)	
	2	1	(1%)	3	(2%)	
	3	1	(1%)	1	(1%)	

Bold values indicate statistically significance, NA; not available. HDR boost = high dose rate brachytherapy boost, EBRT = external beam radiotherapy.

## Data Availability

The data of HDR boost and part of EBRT for this manuscript can be obtained from the public data base [11] and other part of EBRT can be obtained from the author upon reasonable request.

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
