# Peer review of "Radiotherapy for Clinically Localized T3b or T4 Very-High-Risk Prostate Cancer-Role of Dose Escalation Using High-Dose-Rate Brachytherapy Boost or High Dose Intensity Modulated Radiotherapy"

_cancers, 2021, doi:10.3390/cancers13081856_

Round 1
Reviewer 1 Report
The manuscript represents valuable observations on large and unique patient material. However, there are two missing and important issues to be considered.
- The turning point of high-risk (and very high-risk) localized prostate cancer radiotherapy success is the applied dose level. Multiple publications of the recent years (Meta-analyses, Phase III trial long term results, review publications on local dose escalation - whatever the application technology is) show the outcome improvement paired with higher applied dose level. Unfortunately, these papers and information are not involved, whether in the introduction nor the discussion section. All these data support the statistically significant advantage of the application of EBRT+HDR since the combination offers a much higher target dose level (for example, Deutsch et al. 2010= BED 190,08Gy (86,4 Gy IMRT) versus IMRT+HDR resulting in BED 229Gy). Interestingly, Stock et al.2013 published a review of I-125 seed monotherapy results and stated the more favorable treatment outcome if >150 Gy was applied.
- The applied dose in both arms of the presented study is lower (high-dose-IMRT: EQD2 78 Gy versus IMRT+HDR EQD2 108,7Gy) as the observed cut-off applied dose levels of studies stating significant differences between any external beam therapy versus combined external beam + brachytherapy boost.
- Therefore, the authors' conclusion about the equal outcome of high-dose IMRT and EBRT+HDR is an important observation on their particular applied dose level - but it is not true as a general statement. This information needs to be discussed, and the relevant literature analyzed before publication.
- Nevertheless, the manuscript offers key information for finding the real cut-off of local dose escalation in treating high- and very high-risk localized prostate cancer.
Reviewer 2 Report
The aim of the study is of great interest. The essential message of this study is that dose escalation is beneficial for the very high risk group.
However, I suggest that the HDEBRT group and the HDR boost are managed as one group due to too few patients in the HDEBRT group. Furthemore, there is no rationale to present the outcome separately for T3b and T4, as is indicated in the multi-variate analysis in Table3.
I question the selection of patients and in particular the inclusion of patients with very high iPSA. A reasonable cut off would have been an iPSA below 100.
Nothing is reported about the follow up schedule of the included patients.
Specific comments
The very low p-values presented in Table 1b for difference between the 3 dose groups concerning age, iPSA, Gleason score, hormonal therapy are difficult to believe in from the figures presented.
EQD 2 Gy is presented for Conv EBRT, but not for HDEBRT and HDR boost,which also included unconventional fractionation schedules. The alfa/beta ratio used should be given.
Para 3.1. The last sentence is unclear (line140-142). Do you mean that the period of adjuvant hormonal therapy was longer than for the patients treated with EBRT?
Fig.2a. The patient numbers for the Conv- EBRT are omitted.
Line 187: Which endpoint are the figures associated with. I realized that they present PPS.
Line 224: EBRT should be HD EBRT
Discussion: More information about the benefit of dose escalation of the high risk and the expected outcome for this group should be presented including relevant references.
The outcome presented in the paper for the very high risk group is far better than expected. This has to be highlighted as well as potential re reasons.
Also the the GU-toxicity is lower than expected from what has been reported in the literature. This should also be comment upon.
Round 2
Reviewer 1 Report
All reviewer concern were adequately answered.
Author Response
All reviewer concern were adequately answered.
Thank you.
Reviewer 2 Report
I think you have improved the manuscript substantially. The message is now more accurate, and the discussion elucidate the outcome in a satisfactory wa.
Author Response
I think you have improved the manuscript substantially. The message is now more accurate, and the discussion elucidate the outcome in a satisfactory wa.
Thank you.